# DLBCL Cells with Acquired Resistance to Venetoclax Are Not Sensitized to BIRD-2 But Can Be Resensitized to Venetoclax through Bcl-XL Inhibition

**DOI:** 10.3390/biom10071081

**Published:** 2020-07-21

**Authors:** Martijn Kerkhofs, Tamara Vervloessem, Kinga B. Stopa, Victoria M. Smith, Meike Vogler, Geert Bultynck

**Affiliations:** 1Lab. Molecular and Cellular Signaling, Department of Cellular and Molecular Medicine and Leuven Kanker Instituut, Campus Gasthuisberg ON-I, KU Leuven, 3000 Leuven, Belgium; martijn.kerkhofs@kuleuven.be (M.K.); tamara.vervloessem@kuleuven.be (T.V.); 2Malopolska Centre of Biotechnology, Jagiellonian University, 30-387 Krakow, Poland; kinga.stopa@doctoral.uj.edu.pl; 3Department of Molecular and Cell Biology, University of Leicester, Leicester LE1 7RH, UK; vs209@leicester.ac.uk; 4Institute for Experimental Cancer Research in Pediatrics, Goethe-University, 60528 Frankfurt, Germany; m.vogler@kinderkrebsstiftung-frankfurt.de

**Keywords:** Ca^2+^ signaling, ABT-199, BH3 mimetics, Bcl-2, Bcl-XL, IP_3_ receptors

## Abstract

Anti-apoptotic Bcl-2-family members are frequently dysregulated in both blood and solid cancers, contributing to their survival despite ongoing oncogenic stress. Yet, such cancer cells often are highly dependent on Bcl-2 for their survival, a feature that is exploited by so-called BH3-mimetic drugs. Venetoclax (ABT-199) is a selective BH3-mimetic Bcl-2 antagonist that is currently used in the clinic for treatment of chronic lymphocytic leukemia patients. Unfortunately, venetoclax resistance has already emerged in patients, limiting the therapeutic success. Here, we examined strategies to overcome venetoclax resistance. Therefore, we used two diffuse large B-cell lymphoma (DLBCL) cell lines, Riva WT and venetoclax-resistant Riva (VR). The latter was obtained by prolonged culturing in the presence of venetoclax. We report that Riva VR cells did not become more sensitive to BIRD-2, a peptide targeting the Bcl-2 BH4 domain, and established cross-resistance towards BDA-366, a putative BH4-domain antagonist of Bcl-2. However, we found that Bcl-XL, another Bcl-2-family protein, is upregulated in Riva VR, while Mcl-1 expression levels are not different in comparison with Riva WT, hinting towards an increased dependence of Riva VR cells to Bcl-XL. Indeed, Riva VR cells could be resensitized to venetoclax by A-1155463, a selective BH3 mimetic Bcl-XL inhibitor. This is underpinned by siRNA experiments, demonstrating that lowering Bcl-XL-expression levels also augmented the sensitivity of Riva VR cells to venetoclax. Overall, this work demonstrates that Bcl-XL upregulation contributes to acquired resistance of DLBCL cancer cells towards venetoclax and that antagonizing Bcl-XL can resensitize such cells towards venetoclax.

## 1. Introduction

The Bcl-2 protein family regulates the onset of apoptosis by critically controlling mitochondrial outer membrane permeabilization [1,2]. The balance between pro-apoptotic (such as BAX/BAK and BH3-only proteins) and anti-apoptotic (such as Bcl-2, Bcl-XL and Mcl-1) family members is a crucial factor in determining cell death and survival [3]. Anti-apoptotic Bcl-2 protein family members are upregulated in various cancers, e.g., chronic lymphocytic leukemia (CLL) [4], diffuse large B-cell lymphoma (DLBCL) [4], but also solid tumors such as prostate cancer [5]. This enables cancer cells to survive despite pro-apoptotic, oncogenic stress [6]. A crucial feature of anti-apoptotic Bcl-2 family members is their hydrophobic cleft, consisting of Bcl-2 homology (BH) domains 1, 2 and 3 [1]. The hydrophobic groove interacts with the BH3 domain of pro-apoptotic Bcl-2 family members, thereby sequestering them and inhibiting their pro-apoptotic function [1,7]. Venetoclax is the first clinically used BH3 mimetic to treat relapsed CLL cancer patients [8,9,10] and does so by occupying the hydrophobic groove of the anti-apoptotic protein Bcl-2 [7,11]. This prevents the sequestration of pro-apoptotic Bcl-2 family proteins, often upregulated due to oncogenic stress [12] and tips cancer cells “addicted” to Bcl-2 for survival over the edge of apoptosis [13]. Ever since its launch in the clinic, venetoclax has been a precision medicine with great potential and success [11]. While initially it was approved as a therapy for CLL, current research focuses on its use in other types of cancer such as DLBCL, follicular lymphoma, myeloma and acute myeloid leukemia [11,14,15]. 

However, as is often the case with targeted therapies, some cancers developed resistance against venetoclax, rendering it ineffective in long-term treatments to cure patients of cancer [11]. To understand the mechanisms underlying venetoclax resistance better and potentially exploit them, we decided to investigate whether venetoclax-resistant cells become reliant on the antagonizing role of Bcl-2 role at the endoplasmic reticulum (ER) [16,17]. The ER serves as the main intracellular Ca^2+^ storage organelle that controls a plethora of cell functions including cell survival [18]. During the last decades, several Bcl-2-family members have been identified at the ER where they functionally impact Ca^2+^ homeostasis and dynamics [19,20]. Bcl-2 directly targets IP_3_Rs via its BH4 domain and inhibits Ca^2+^ flux through the channel, preventing apoptosis [21,22,23]. Targeting Bcl-2 at the ER can be achieved through Bcl-2-IP_3_R disrupter 2 (BIRD-2) [24,25,26]. This is a decoy peptide for Bcl-2 based on its binding sequence in the central, regulatory domain of the inositol 1,4,5-trisphosphate receptor (IP_3_R) [17,23,27]. BIRD-2 can kill several Bcl-2-dependent cancer cell types, including primary CLL cells and DLBCL cell lines, by provoking intracellular Ca^2+^ overload [26,27,28]. At the molecular level, we found that the combined upregulation of the type 2 IP_3_R (IP_3_R2 isoform) in combination with ongoing chronic IP_3_ signaling downstream of the chronically/tonically active B-cell receptor rendered cancer cells sensitive to BIRD-2 exposure [28,29].

Moreover, previous work from our team revealed that there is a reciprocal correlation between venetoclax sensitivity and BIRD-2 sensitivity, i.e., cancer cells that were more sensitive to venetoclax were more resistant to BIRD-2 and vice versa [30]. Hence, in this work, we wished to examine whether cancer cells that have become resistant to venetoclax were more sensitive to BIRD-2. Therefore, we compared BIRD-2 sensitivity in an isogenic cell model with acquired venetoclax resistance. This was achieved by culturing the DLBCL cell line Riva in conditions of chronic exposure to venetoclax. Here, we compared the sensitivity towards BIRD-2 between naïve Riva cells (Riva WT) and isogenic Riva cells with acquired resistance towards venetoclax (Riva VR).

We reported that Riva VR do not develop increased sensitivity towards BIRD-2 compared to their naïve counterparts. Consistently, we did not see an upregulation of the IP_3_R. In addition, we found that Bcl-XL upregulation is at least in part responsible for the resistance to venetoclax, while Mcl-1-expression levels appeared unaltered. Finally, we observed cross-resistance towards BDA-366, a molecule proposedly targeting the BH4 domain of Bcl-2 [31].

## 2. Materials and Methods

### 2.1. Cell Culture and Chemicals

Riva WT and VR were grown in RPMI (Gibco/Invitrogen, Merelbeke, Belgium) supplemented with 2 mM glutamax, 10% heat-inactivated fetal bovine serum, 100 IU/mL penicillin and 100 µg/mL streptomycin (100× Pen/Strep, Gibco/Invitrogen, Merelbeke, Belgium) at 37 °C and 5% CO_2_. Resistance to venetoclax was achieved by culturing Riva cells in increasing concentrations of venetoclax over more than three months. This selection started with low nanomolar concentrations of venetoclax (1–10 nM) and concentrations were stepwise increased up to 300 nM when more than 30% of cells survived the selection. Survival of cells was monitored using AnnexinV-FITC staining and flow cytometry twice a week. Venetoclax was purchased from Active Biochem (Kowloon, Hong Kong). Cells were seeded 24 h before being used in experiments at a density of 250,000/mL. BIRD-2 peptide (sequence: RKKRRQRRRGGNVYTEIKCNSLLPLAAIVRV) was purchased from LifeTein (South Plainfield, NJ, USA) with a purity of >85%. BDA-366 was purchased from BioVision (Milpitas, CA, USA). Both S63845 and A-1155463 were purchased from Selleckchem (Munich, Germany).

### 2.2. Flow Cytometry: Apoptosis Assay

Riva WT and VR cells were treated for 24 h with venetoclax, BIRD-2, BDA-366, S63845 or A-1155463 or a combination of these. Subsequently, cells were pelleted by centrifugation and incubated with AnnexinV-FITC (Life Technologies, Carlsbad, CA V13245, USA,) and 7-AAD (Becton Dickinson, Franklin Lakes, NJ 555815, USA). Cell suspensions were analyzed with an Attune Acoustic Focusing Flow Cytometer (Applied Biosystems). Cell death by apoptosis was scored by quantifying the population of AnnexinV-FITC-positive cells using FlowJo version 10 software.

### 2.3. Western Blot

Riva WT and VR cells were washed with phosphate-buffered saline and incubated at 4 °C with lysis buffer (20 mM Tris-HCl (pH 7.5), 150 mM NaCl, 1.5 mM MgCl_2_, 0.5 mM dithiothreitol, 1% Triton-X-100 and 1 tablet of complete EDTA-free protease inhibitor (Thermo Scientific, Brussels, Belgium)) for 30 min. Cell lysates were centrifuged for 5 min at 4000× *g* and analyzed by Western blotting as previously described [21]. Western blot quantification was done using Image Lab 5.2 software (Bio-Rad Laboratories, Temse, Belgium).

### 2.4. Intracellular Ca^2+^ Measurement in Intact Cells

Riva VR and WT cells were loaded with 1 µM Fura-2-AM (Eurogentec, Seraing, Belgium) at room temperature in modified Krebs solution (150 mM NaCl, 5.9 mM KCl, 1.2 mM MgCl_2_, 11.6 mM HEPES (pH 7.3), 11.5 mM glucose and 1.5 mM CaCl_2_) for 30 min. This was followed by a de-esterification step in the absence of extracellular Fura-2-AM for 30 min at room temperature. Extracellular Ca^2+^ was chelated with EGTA before stimulating cells with IgG/IgM (12 μg/mL; Jackson ImmunoResearch Europe Ltd., Cambridge, UK) to elicit intracellular Ca^2+^ signaling. Alternatively, thapsigargin (1 μM), an inhibitor of SERCA, was used to deplete the ER to gauge ER Ca^2+^ content. Fluorescence was monitored using a Flexstation 3 microplate reader (Molecular Devices, Sunnyvale, CA, USA) by alternating the excitation of Fura-2 at 340 and 380 nm and collecting the emission at 510 nm. All traces are shown as the ratio of emitted fluorescence of Fura-2 (F340/F380). GraphPad Prism 8 was used to calculate area under the curve (AUC). 

### 2.5. siRNA Transfection of Riva VR Cells

Riva VR cells were transfected using the Amaxa^®^ Cell Line Nucleofector^®^ Kit L (Lonza, Basel, Switzerland), program C-05, as described in Bittremieux et al. [29]. Briefly, 3 × 10^6^ cells were transfected with 500 nM siCTRL (ON-TARGET plus, non-targeting control pool, from Dharmacon) and 500 nM siBcl-XL (hs.Ri.BCL2L1.13.1, from IDT). At 24 h post-transfection, the cells were used for experiments and collected for Western blot analysis to confirm knockdown of Bcl-XL. 

## 3. Results

### 3.1. Characterization of the Venetoclax-Sensitive and -Resistant Riva Cells

To obtain venetoclax-resistant Riva cells (Riva VR), parental Riva cells (Riva WT) were chronically exposed to increasing concentrations of venetoclax. A dose–response experiment indicated approximately a tenfold difference in venetoclax sensitivity as demonstrated by the different EC_50_ values for venetoclax (Figure 1a,b) derived from FACS measurements (AnnexinV-7-AAD staining). Thus, these data confirm the presence of venetoclax resistance in the VR cell line relative to the parental cell line.

### 3.2. Acquired Venetoclax Resistance does not Induce Increased Sensitivity Towards BIRD-2 

Previous work performed by our lab [30] revealed an inverse correlation between venetoclax and BIRD-2 sensitivity in DLBCL cell lines. Based on these findings, we hypothesized that the cells with acquired resistance to venetoclax could have become more susceptible to BIRD-2, illustrating a shift from Bcl-2’s reliance on the hydrophobic cleft towards a BH4 domain-dependent mechanism. However, a dose–response experiment showed no significant difference in the EC_50_ values of BIRD-2 in Riva VR compared to Riva WT (Figure 2a). As in previous work where we associated BIRD-2 sensitivity of DLBCL cells with IP_3_R2-expression levels [28], we compared the expression of IP_3_R2 and the other IP_3_R isoforms between Riva WT and Riva VR. Yet, consistent with the findings that Riva VR cells were not more sensitive to BIRD-2 than Riva WT cells, IP_3_R2-expression levels were very similar between Riva WT and Riva VR cells (Figure 2b). Similarly, the expression levels of the other isoforms of the IP_3_R (IP_3_R1 and -3) were not different between Riva WT and Riva VR cells (Figure 2b). These data demonstrate that cancer cells with acquired venetoclax resistance do not become dependent on Bcl-2’s non-canonical role at the ER for survival.

### 3.3. IgG/IgM-Induced Cytosolic Ca^2+^ Signaling and ER Store Content Do not Differ between Riva WT and Riva VR 

Since we observed that IP_3_R levels were unchanged, we wondered whether Ca^2+^ handling in Riva WT and Riva VR was also unaltered. To this end, we conducted cytosolic Ca^2+^ measurements by loading the cells with Fura-2-AM. Cytosolic Ca^2+^ signals were triggered by the addition of IgG/IgM (12 μg/mL) in the presence of 3 mM EGTA, an extracellular Ca^2+^-chelating agent (Figure 3a). The latter ensures that only Ca^2+^ release from intracellular Ca^2+^ stores occurs without Ca^2+^ influx. The cytosolic Ca^2+^ signals elicited by B-cell receptor stimulation via IgG/IgM were very similar between Riva WT and Riva VR, as is evident from the area under the curve (AUC) (Figure 3b). Furthermore, we used thapsigargin (1 μM) to indirectly quantify the ER Ca^2+^ store content. However, no significant difference was observed in the thapsigargin-releasable Ca^2+^ pool between Riva WT and Riva VR, indicating that Ca^2+^ levels present in the ER of Riva VR cells and of Riva WT were very similar (Figure 3c). 

### 3.4. Acquired Venetoclax Resistance Is Associated with Bcl-XL Upregulation and Bcl-XL Dependence

Next, we decided to check the levels of selected Bcl-2 protein family members via Western blot, since they are known to be involved in venetoclax resistance [32,33]. While Mcl-1 levels were similar in Riva WT and Riva VR cells, a significant increase in Bcl-XL-protein levels was observed in Riva VR in the Western blots (Figure 4a). To investigate whether Bcl-XL upregulation was indeed responsible for the induced venetoclax resistance, we treated both Riva WT and Riva VR cells with a panel of validated on-target BH3-mimetic Bcl-2-family inhibitors: venetoclax (10 nM and 300 nM), a selective Bcl-XL inhibitor A-1155463 (1 μM and 3 μM) and a selective Mcl-1 inhibitor S63845 (1 μM and 10 μM) [34]. As confirmed before, Riva WT and Riva VR displayed a large difference in sensitivity towards venetoclax (Figure 4b). Excitingly, the venetoclax resistance could be overcome by simultaneously treating the cells with venetoclax (10 nM) and the Bcl-XL inhibitor A-1155463 (1 μM). It is interesting to note that 1 μM of A-1155463 did not provoke cell death when administered alone. Additionally, both Riva WT and VR were sensitive to this compound at higher concentrations (3 μM), pointing to a pre-existing susceptibility to the inhibition of Bcl-XL. However, Bcl-XL inhibition by itself was much less potent in killing the cells than combined Bcl-XL/Bcl-2 inhibition. This illustrates that even though Riva VR cells seemed to become more dependent on Bcl-XL when treated with venetoclax, they did not lose the ability to block cell death via Bcl-2. Inhibition of Mcl-1’s hydrophobic cleft did not restore venetoclax sensitivity (Figure 4b). These data may indicate that Bcl-XL upregulation contributes to the survival of Riva cells after chronic exposure to venetoclax. Furthermore, this illustrates that Riva VR cells achieve a status of dual dependency on Bcl-2 and Bcl-XL, but not on Mcl-1.

### 3.5. Knockdown of Bcl-XL Resensitizes Riva VR Cells to Venetoclax

To further corroborate these findings, we used a siRNA-mediated knockdown approach to decrease the expression levels of Bcl-XL in Riva VR cells. Western blot analysis validated that siBcl-XL could reduce Bcl-XL-protein levels in Riva VR cells (Figure 5a). Furthermore, cell death analysis in Riva VR cells treated with siCTRL or siBcl-XL revealed that knockdown of Bcl-XL sensitized the cells to venetoclax (300 nM) (Figure 5b). In addition, Riva WT and VR cells were treated with ABT-737, a non-selective BH3 mimetic inhibitor of both Bcl-2 and Bcl-XL [35]. While Riva VR cells are much more resistant to venetoclax than Riva WT cells, Riva VR and Riva WT seemed equally sensitive to ABT-737 (Figure 5c). Both experiments underpin our hypothesis that the resistance of Riva VR cells towards venetolax is at least in part due to upregulation of Bcl-XL.

### 3.6. Acquired Venetoclax Resistance in Riva Cells Generates Cross-Resistance for BDA-366

Finally, we also assessed the sensitivity towards another presumed BH4-domain inhibitor of Bcl-2, BDA-366 [31]. This small molecule was identified from a screen for BH4-domain-interacting molecules and is thought to switch Bcl-2 into a pro-apoptotic protein by triggering surface exposure of its BH3 domain [36]. However, recent work has cast some doubts on whether BDA-366 is a bona fide Bcl-2 inhibitor [34]. Indeed, when Riva WT and Riva VR cells were treated with BDA-366 in a dose–response experiment, Riva WT cells were approximately ten times more sensitive to the compound than Riva VR cells (Figure 6). This indicates that induction of venetoclax resistance also results in resistance against BDA-366.

## 4. Discussion

The launch of venetoclax in the clinic to combat CLL has been a relative success [11]. While this molecule improved the treatment of patients with CLL, some patients develop resistance against venetoclax very quickly [11]. Thus, it is important to elucidate the mechanisms underlying venetoclax resistance and to develop strategies for killing such venetoclax-resistant cancer cells. To this end, we compared the DLBCL cell line Riva before and after chronic venetoclax exposure. By treating the cells for a longer time with venetoclax, they become resistant to it. In this way we are able to study the mechanisms involved in venetoclax resistance. 

As we previously showed that there was a reciprocal sensitivity among a collection of DLBCL cells between BIRD-2 and venetoclax [30], we were interested to assess whether DLBCL cells with acquired resistance towards venetoclax could have become sensitized towards BIRD-2. However, our data revealed that venetoclax resistance did not shift Bcl-2’s anti-apoptotic function from dependency on the hydrophobic cleft to using its BH4 domain to suppress cell death. Indeed, both cell lines displayed similar sensitivities to BIRD-2, a BH4 domain targeting peptide, and Riva VR did not show an upregulation of IP_3_R2-protein levels, a parameter shown to correlate with BIRD-2 sensitivity in previous work [28]. We did not observe any differences in IP_3_R expression levels and neither in IgG/IgM-induced cytosolic Ca^2+^ signaling. 

Instead, we observed an upregulation of anti-apoptotic Bcl-2 family member Bcl-XL in Riva VR cells compared to regular Riva cells. We demonstrated that this Bcl-XL upregulation contributed to venetoclax resistance, since siRNA-mediated knockdown of Bcl-XL in Riva VR cells could sensitize these cells to venetoclax. Consequently, the venetoclax resistance of Riva VR cells could be overcome by validated BH3-mimetic drugs [34], such as A-1155463, a selective antagonist of Bcl-XL or ABT-737, a non-selective Bcl-2/Bcl-XL inhibitor. Upregulation of Bcl-XL, in cooperation with Mcl-1, is a resistance mechanism that has previously been reported for other DLBCL cell models, such as SU-DHL-6, OCI-LY-19 and OCI-LY-1 [33,37]. Venetoclax resistance in OCI-LY-1 cells (as well as CLL patient cells) was also accompanied by a remodeling of the mitochondrial metabolism involving activation of AMPK and increased OXPHOS [33]. 

Interestingly, Riva VR cells seem to develop a dual dependency on Bcl-2 family members, since inhibiting Bcl-XL alone is not sufficient to completely kill Riva VR cells. This shows that while Bcl-XL upregulation may contribute to acquiring resistance against venetoclax, Riva VR cells do not shift away from using Bcl-2 as an anti-apoptotic buffer. It has been reported that different members of the Bcl-2 protein family have different affinities for their pro-apoptotic counterpart [38]. Yet, functional redundancy of Bcl-2-protein family members has been described [39,40]. Our findings are in line with this concept, rendering it more challenging to target a single Bcl-2 family member for prolonged treatment of cancer due to compensatory upregulation of other Bcl-2 family members providing drug resistance. In this study, we did not examine the mechanism by which Bcl-XL is upregulated in Riva VR. A previous study in B-cell lymphoid cell models, however, revealed downregulation of miR-377, a microRNA suppressing Bcl-XL expression, in venetoclax-resistant cell models [41].

Furthermore, our data suggest that venetoclax resistance may generate cross-resistance to other molecules that presumably target Bcl-2, such as BDA-366. BDA-366 is proposed to target Bcl-2’s BH4 domain and turn Bcl-2 into a pro-apoptotic protein [31,36]. Thus, while BDA-366 and venetoclax inhibit Bcl-2 via distinct mechanisms, cancer cells that acquire resistance against venetoclax also seem to acquire resistance against BDA-366. Yet, it should be noted that recent work of the Letai group has questioned BDA-366 as an on-target inhibitor of Bcl-2 proteins [34], indicating that BDA-366 can kill cancer cells independently of Bcl-2. Thus, the results obtained with BDA-366 may need to be interpreted with care in relation to BH4-domain antagonism.

Different mechanisms for venetoclax resistance have been described in different types of cancer [32,37,42,43,44]. Beyond upregulation of other anti-apoptotic Bcl-2-family members that are not inhibited by venetoclax, Bcl-2 mutations have emerged in CLL patients undergoing venetoclax treatment. A paired analysis of pre-venetoclax and venetoclax progression in 15 CLL patients revealed a de novo Bcl-2 mutation, changing residue Gly101 in Val in 7 CLL patients [32]. Bcl-2^G101V^ displayed a severely reduced affinity for binding venetoclax. This rendered the drug inefficient in disrupting Bcl-2’s association with pro-apoptotic family members and enabled cancer cell resistance towards venetoclax. In follow-up work, it was shown that Bcl-2 mutations were frequently occurring in CLL patients receiving venetoclax, including mutations at other residues such as Asp103Tyr [42]. More recently, several other Bcl-2 mutations have been identified to co-occur with Gly101Val, whereby several patients accumulated multiple Bcl-2 mutations [45]. From the latter study, it is clear that different venetoclax-resistance mechanisms can arise, including the upregulation of Bcl-XL. In addition, cancer cells may remodel signaling pathways to elicit resistance against chemotherapy. For example, upregulation of Akt activity was associated with venetoclax resistance and severely reduced Bcl-2 and Bim expression levels [46]. Interestingly, in the same vein, kinase inhibitors were found to be effective to overcome venetoclax resistance, possibly interfering with kinase-mediated responses and subsequent remodeling caused by signaling cues from the microenvironment [47]. A similar mechanism was found in primary patient samples of DLBCL [48].

On a more profound level, it is interesting to ask whether certain cancer cells are “primed” to follow a predetermined path towards resistance. If that were the case, would Riva cells consequently upregulate Bcl-XL in response to chronic venetoclax exposure, or would a different resistance mechanism develop? There may be factors underlying this response, and identification of these factors would allow for a prediction of possible venetoclax resistance in patients. If the resistance occurs by coincidence, it is important to identify the parameters setting out the path towards resistance. 

## 5. Conclusions

Venetoclax resistance has emerged as a complicating factor for the prolonged treatment of CLL patients with the drug, limiting long-term therapeutic success in the clinic. Therefore, strategies to overcome cancer cell resistance to venetoclax are highly needed. Since venetoclax is investigated as a potential drug against DLBCL, we anticipate similar problems in the treatment of these patients. Using an isogenic DLBCL cell model with high and low venetoclax sensitivity, we found that Bcl-XL is upregulated in venetoclax-resistant cells, while the cells do not become more sensitive to BIRD-2, a peptide that disrupts IP_3_R/Bcl-2 complexes and disrupts Bcl-2’s function at the ER. Similarly, venetoclax resistance seems to induce cross-resistance towards BDA-366, a small molecule reported to target the Bcl-2 BH4 domain. Therefore, Bcl-XL upregulation is a factor that cancer cells exploit to acquire resistance towards venetoclax and which may be seized upon to improve the treatment of cancer patients that lose responsiveness towards venetoclax treatment. 

## Figures and Tables

**Figure 1 biomolecules-10-01081-f001:**
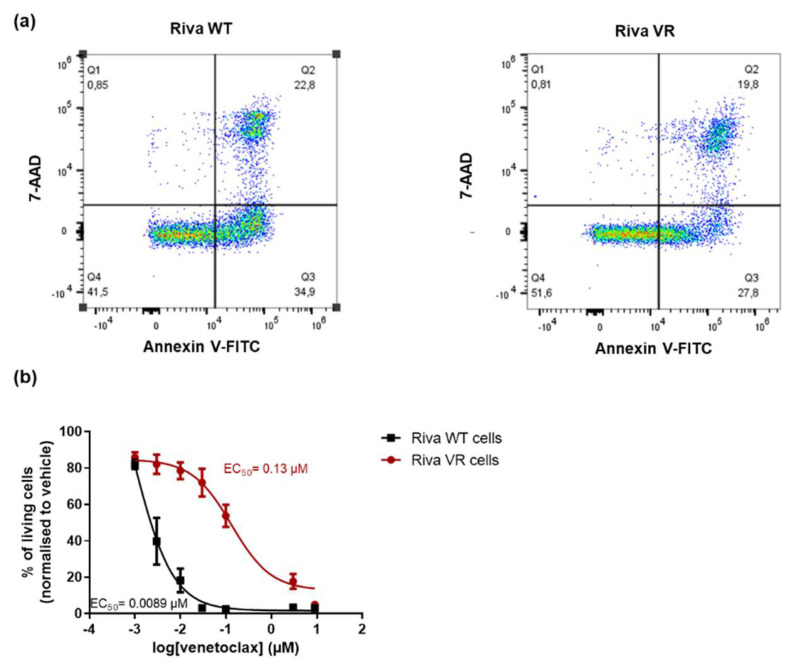
Riva VR cells are resistant to venetoclax as compared to Riva WT cells. (**a**) Representative dot plots from flow cytometric analysis of AnnexinV-FITC/7-AAD stained Riva wild-type (WT) and venetoclax-resistant (VR) cells, treated with venetoclax at 3 nM and 100 nM, respectively, during 24 h. (**b**) Dose–response curves of Riva WT and Riva VR 24 h after drug exposure. The apoptotic population was defined as the AnnexinV-FITC-positive fraction. Data presented are average ± SEM (*n* = 5).

**Figure 2 biomolecules-10-01081-f002:**
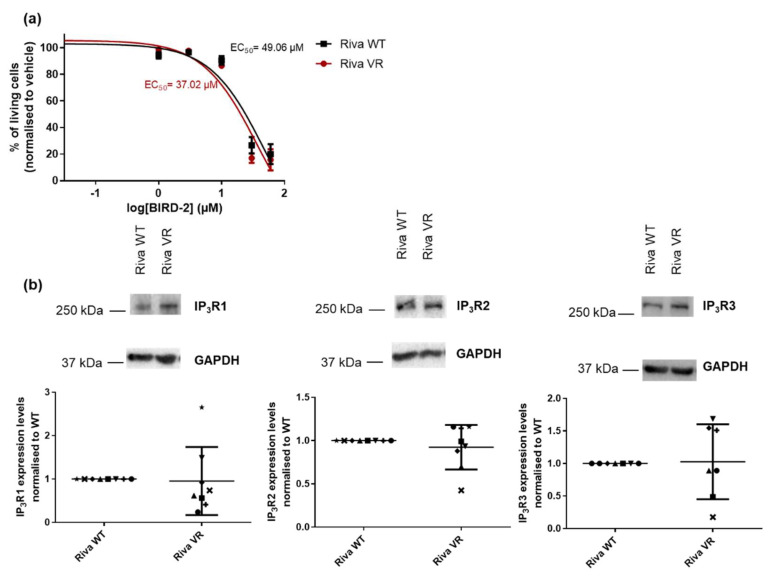
Riva WT and VR cells have a similar sensitivity towards BIRD-2 treatment. (**a**) Dose–response curves of Riva WT and VR cells after incubation with BIRD-2 for 24 h. The apoptotic population was defined as the AnnexinV-FITC-positive fraction. Data presented are average ± SEM (*n* = 5). (**b**) Representative Western blots of IP_3_R1, 2 and 3-expression levels in Riva WT and Riva VR cells. GAPDH was used as a loading control. Quantification of IP_3_R1, -2 and -3-protein levels in Riva WT and Riva VR cells. For each blot, the immunoreactive bands were quantified and normalized towards the signal obtained for Riva WT cells, which was set at 1. Data are presented as average ± SD (*n* ≥ 7).

**Figure 3 biomolecules-10-01081-f003:**
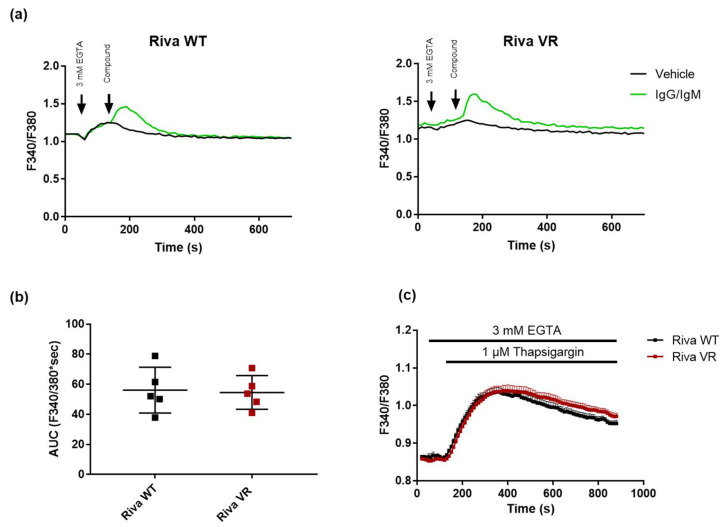
Cytosolic IgG/IgM Ca^2+^ responses and ER Ca^2+^ content do not differ between Riva WT and Riva VR. (**a**) Typical cytosolic Ca^2+^ traces in Fura-2-AM-loaded Riva WT and VR. Sixty seconds after the chelation of extracellular Ca^2+^ with EGTA (3 mM), the cells were stimulated with vehicle or 12 μg/mL IgG/IgM to provoke a cytosolic Ca^2+^ signal. (**b**) Analysis of the area under the curve (AUC) of the IgG/IgM-triggered peak. Data are presented as average ± SEM (N = 5). (**c**) Cytosolic Ca^2+^ traces in Fura-2-AM-loaded Riva WT (black) and VR (red). Sixty seconds after the chelation of extracellular Ca^2+^ with EGTA (3 mM), the cells were treated with thapsigargin (1 μM) to deplete ER Ca^2+^ stores. Data are presented as average ± SEM (N = 6).

**Figure 4 biomolecules-10-01081-f004:**
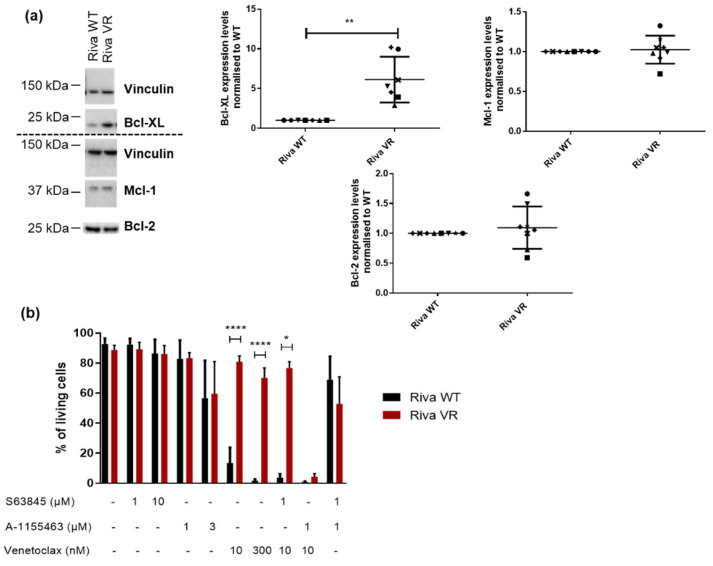
Bcl-XL-expression levels are upregulated in Riva VR cells. (**a**) Left: representative Western blots of Bcl-XL-, Mcl-1- and Bcl-2-expression levels in Riva WT and VR cells. Vinculin was used as a loading control. Right: quantification of Bcl-XL-, Mcl-1- and Bcl-2-expression levels. Expression levels in Riva VR were normalized on Riva WT, which was set at 1. Data are presented as the average ± SD (*n* ≥ 7); ** *p* < 0.01. (**b**) Flow cytometric analysis of AnnexinV-FITC negative and 7-AAD negative cells treated with BH3-mimetic inhibitors of Bcl-XL (A-1155463), Mcl-1 (S63845) and Bcl-2 (venetoclax) for 24 h. Data are represented as mean ± SD (*n* = 5); * *p* < 0.05, **** *p* < 0.001.

**Figure 5 biomolecules-10-01081-f005:**
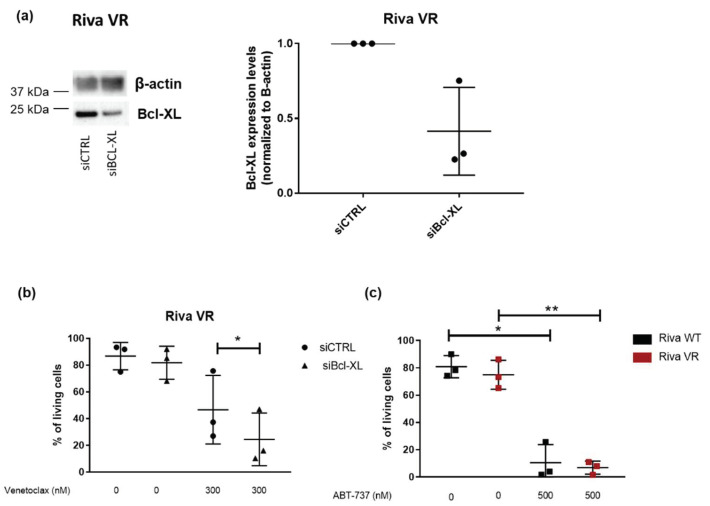
Bcl-XL knockdown sensitizes Riva VR cells to venetoclax. (**a**) Left: representative Western blots of Bcl-XL-expression levels in Riva VR cells after treatment with siCTRL or siBcl-XL. β-actin was used as a loading control. Right: quantification of Bcl-XL-expression levels. Expression levels in Riva VR were normalized to the Bcl-XL levels in siCTRL-treated cells, which were set at 1. Data are represented as the average ± SD (*n* = 3). (**b**) Flow cytometric analysis of AnnexinV-FITC negative and 7-AAD negative cells treated with venetoclax for 24 h. Data are ± SD (N = 3); * *p* < 0.05. (**c**) Flow cytometric analysis of AnnexinV-FITC negative and 7-AAD negative Riva WT and VR cells treated with ABT-737 for 24 h. Data are ± SD (*n* = 3); * *p* < 0.05, ** *p* < 0.01.

**Figure 6 biomolecules-10-01081-f006:**
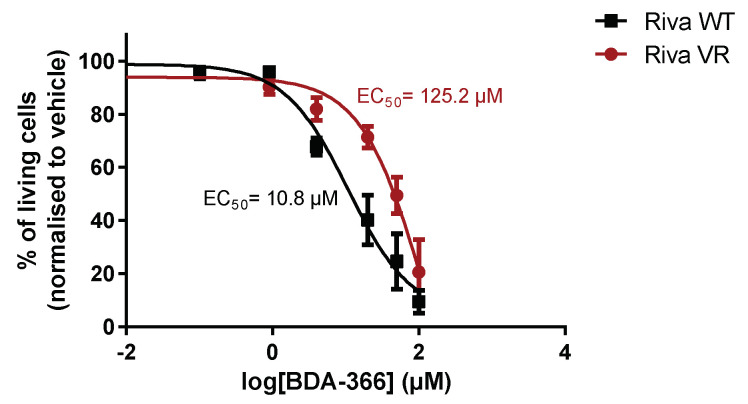
Venetoclax resistance induces cross-resistance towards BDA-366. Dose–response curves of Riva WT and Riva VR after incubation with BDA-366 for 24 h. The apoptotic population was defined as the AnnexinV-FITC-positive fraction. Data represented as average ± SEM (*n* = 4).

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
