# Peer review of "DLBCL Cells with Acquired Resistance to Venetoclax Are Not Sensitized to BIRD-2 But Can Be Resensitized to Venetoclax through Bcl-XL Inhibition"

_biomolecules, 2020, doi:10.3390/biom10071081_

Round 1
Reviewer 1 Report
The manuscript submitted by Martijn et al. deals with elucidation of an acquired resistance of DLBCL cell line Riva to BH3 mimetic drug venetoclax, which targets the anti-apoptotic protein Bcl-2. The authors show that the prolonged exposure of Riva cells to increasing venetoclax (VTX) concentration led to the selection of Riva-VTX-resistant clone (over an order of magnitude) and looked for possible mechanisms of this acquired resistance to VTX. These cells remained similarly sensitive to BIRD-2, a peptide targeting Bcl-2-IP3R interaction via Bcl-2 BH4 domain, which causes release of ER calcium stores and cell death due to calcium overload. Interestingly, BDA-366, another reported agent blocking BH4 domain of Bcl-2 was in analogy with venetoclax significantly less effective towards Riva-VTX-R cells. Examining expression of other major anti-apoptotic Bcl-2 family proteins, the authors uncovered that the increased expression of Bcl-Xl (and interestingly not Mcl-1) might be behind enhanced resistance of these cells to VTX (and likely BDA-366). This finding might represent another mode of the acquired VTX resistance (in addition to Bcl-2 mutations and increased Mcl-1 expression), though its path-physiological relevance is questionable.
Comments and notes:
- Is the increased expression of Bcl-Xl (Fig. 4a – labeled as Fig. 1 – likely typo) due to its enhanced transcription or post-translational stabilization? Also, though the authors claim that Bcl-2 expression in the VTX-resistant Riva cells remains unchanged, Fig.4a shows a substantial increase in its expression as well.
- To prove involvement of Bcl-Xl, it would be instrumental to also suppress its expression (in addition to pharmaceutic inhibition by A-1155463) using siRNA and thus revert VTX resistance as well.
- ABT-737/novitoclax targets in addition to Bcl-2 also Bcl-Xl and Bcl-w and therefore should overcome VTX resistance in these Riva cells – worth of examining.
- In Fig. 3 is stated “chelatation of intracellular Ca2+ with EGTA” – again likely typo – should be extracellular.
- For how long were Riva cells treated and with what concentrations of VTX during the process of the selection of VTX-resistant Riva cells culture?
Author Response
The manuscript submitted by Kerkhofs et al. deals with elucidation of an acquired resistance of DLBCL cell line Riva to BH3 mimetic drug venetoclax, which targets the anti-apoptotic protein Bcl-2. The authors show that the prolonged exposure of Riva cells to increasing venetoclax (VTX) concentration led to the selection of Riva-VTX-resistant clone (over an order of magnitude) and looked for possible mechanisms of this acquired resistance to VTX. These cells remained similarly sensitive to BIRD-2, a peptide targeting Bcl-2-IP3R interaction via Bcl-2 BH4 domain, which causes release of ER calcium stores and cell death due to calcium overload. Interestingly, BDA-366, another reported agent blocking BH4 domain of Bcl-2 was in analogy with venetoclax significantly less effective towards Riva-VTX-R cells. Examining expression of other major anti-apoptotic Bcl-2 family proteins, the authors uncovered that the increased expression of Bcl-Xl (and interestingly not Mcl-1) might be behind enhanced resistance of these cells to VTX (and likely BDA-366). This finding might represent another mode of the acquired VTX resistance (in addition to Bcl-2 mutations and increased Mcl-1 expression), although its pathophysiological relevance is questionable.
Answer: We thank the reviewer for the constructive feedback.
- Is the increased expression of Bcl-Xl (Fig. 4a – labeled as Fig. 1 – likely typo) due to its enhanced transcription or post-translational stabilization? Also, though the authors claim that Bcl-2 expression in the VTX-resistant Riva cells remains unchanged, Fig.4a shows a substantial increase in its expression as well.
Answer: The mechanisms underlying increased expression of Bcl-XL in Riva VR cells was not further investigated. Yet, we discuss this aspect in more detail on Line 297 by a previous study in venetoclax-resistant cell models, reporting downregulation of miR-377, a microRNA targeting Bcl-XL mRNA, in venetoclax-resistant cell models. We now also provide a more representative immunoblot in Fig. 4a that is in line with the quantified levels among different blots. - To prove involvement of Bcl-Xl, it would be instrumental to also suppress its expression (in addition to pharmaceutic inhibition by A-1155463) using siRNA and thus revert VTX resistance as well.
Answer: We appreciate the suggestion of the reviewer. We performed this experiment, demonstrating that venetoclax-resistant cells in which Bcl-XL is knocked down become sensitized to venetoclax. The results have been integrated and discussed (Fig. 5a and Fig. 5b).
- ABT-737/navitoclax targets in addition to Bcl-2 also Bcl-Xl and Bcl-w and therefore should overcome VTX resistance in these Riva cells – worth of examining.
Answer: We appreciate the suggestion of the reviewer. We performed this experiment, demonstrating that ABT-737 indeed can overcome venetoclax resistance in the Riva cells (Fig. 5c).
- In Fig. 3 is stated “chelatation of intracellular Ca2+ with EGTA” – again likely typo – should be extracellular.
Answer: We corrected the typo and indeed this should be extracellular.
- For how long were Riva cells treated and with what concentrations of VTX during the process of the selection of VTX-resistant Riva cells culture?
Answer: We apologize that this information was not provided in the first version of the manuscript. This information is now provided on Page 2 line 92 – Page 3 95.
Reviewer 2 Report
In one DLBCL cell line model the authors show that Bcl-Xl upregulation is the mechanism of resistance to venetoclax. Compounds that target Bcl-2 via different binding domains do not overcome resistance to a Bcl-2 BH3-binding inhibitor.
This mechanism of resistance to Bcl-2 inhibitors is widely known in the apoptosis world, with much evidence from venetoclax in the clinic (more of these references need to be included in both the introduction and discussion sections).
The methods section needs more detail:
-define "prolonged periods" to create Riva-RS cells - weeks? months? any time off drug given during this development? When was drug removed prior to the experiments performed?
-the stepwise methods for the calcium methods are unclear. They are better described in the results section, but need to be clear in the methods section as well. Include any software used to calculate AUCs
-what software was used for western blot quantitation?
Figure 1a legend - during 24 h or after 24 h drug exposure
Figure 2 - suggest making all western blot boxes the same size as viewing the same proteins. Unless looking for additional bands or band shifts there is no need for different sized crop boxes
Figure 3b and c - legend does not clearly identify the red vs black data, nor the square vs triangle data
Figure 4 -labeled as Figure 1! in legend
Figure 4a - Bcl-2 quantitation data suggests no change in Bcl-2 protein upon Ven-resistance but the representative blot of Bcl-2 shows an increase. Perhaps a better representative of the data could be used
Similarly and also were used in conjunction (line 145)
Justify concentrations used for Bcl-Xl and Mcl-1 inhibitors. Why are dose responses +/- venetoclax not shown? This would be preferred data to show range of other inhibitors in presence of the Bcl-2 inhibitor. (Also this EC50 data is shown for other inhibitors in other figures)
IC50 vs EC50 - data shown is EC50 (drug killing half of cells aka response), IC50 is inhibition of a target binding
Paragraph starting at line 277 is muddled and confusing. The goal is to ask the determinant for which mechanism of resistance is utilized by cells. This could be more clearly written.
Discussion - needs more references for venetoclax resistance in clinic for several cancer types including your DLBCL of interest. The concept of Bcl2 family overlap and redundancy is not new, also needs references.
Conclusions - be sure to highlight your specific problem you addressed(DLBCL resistance to venetoclax, not CLL as it is not your model) and the relevance of the model used.
Overall, clearly written, albeit too simplified in areas. Superlatives are used too often (e.g. interestingly, fascinatingly).
Author Response
In one DLBCL cell line model the authors show that Bcl-Xl upregulation is the mechanism of resistance to venetoclax. Compounds that target Bcl-2 via different binding domains do not overcome resistance to a Bcl-2 BH3-binding inhibitor.
Answer: We thank the reviewer for the constructive feedback.
This mechanism of resistance to Bcl-2 inhibitors is widely known in the apoptosis world, with much evidence from venetoclax in the clinic (more of these references need to be included in both the introduction and discussion sections).
Answer: We provided additional references in the discussion.
The methods section needs more detail:
-define "prolonged periods" to create Riva-RS cells - weeks? months? any time off drug given during this development? When was drug removed prior to the experiments performed?
-the stepwise methods for the calcium methods are unclear. They are better described in the results section, but need to be clear in the methods section as well. Include any software used to calculate AUCs
-what software was used for western blot quantitation?
Answer: We now provide this information in the relevant M&M sections.
Figure 1a legend - during 24 h or after 24 h drug exposure
Figure 2 - suggest making all western blot boxes the same size as viewing the same proteins. Unless looking for additional bands or band shifts there is no need for different sized crop boxes
Figure 3b and c - legend does not clearly identify the red vs black data, nor the square vs triangle data
Figure 4 -labeled as Figure 1! in legend
Figure 4a - Bcl-2 quantitation data suggests no change in Bcl-2 protein upon Ven-resistance but the representative blot of Bcl-2 shows an increase. Perhaps a better representative of the data could be used
Similarly and also were used in conjunction (line 145)
Answer: We have updated the legends and also provide more representative immunoblots.
Justify concentrations used for Bcl-Xl and Mcl-1 inhibitors. Why are dose responses +/- venetoclax not shown? This would be preferred data to show range of other inhibitors in presence of the Bcl-2 inhibitor. (Also this EC50 data is shown for other inhibitors in other figures)
Answer: We have used concentrations of Bcl-XL and Mcl-1 inhibitors that by themselves were not sufficient to provoke cell death in the Riva (or Riva VR cells). We then assessed whether these inhibitors at these exact same concentrations could kill Riva VR cells when combined with a low venetoclax concentration that by itself was not sufficient to kill Riva VR cells. This regiment allowed us to investigate the mechanisms by which Riva VR cells acquired venetoclax resistance and to overcome it.
IC50 vs EC50 - data shown is EC50 (drug killing half of cells aka response), IC50 is inhibition of a target binding
Answer: We have changed IC50 to EC50.
Paragraph starting at line 277 is muddled and confusing. The goal is to ask the determinant for which mechanism of resistance is utilized by cells. This could be more clearly written.
Answer: We updated the paragraph.
Discussion - needs more references for venetoclax resistance in clinic for several cancer types including your DLBCL of interest. The concept of Bcl2 family overlap and redundancy is not new, also needs references.
Conclusions - be sure to highlight your specific problem you addressed (DLBCL resistance to venetoclax, not CLL as it is not your model) and the relevance of the model used.
Answer: We seriously revised and expanded the discussion and implemented several additional studies.
Overall, clearly written, albeit too simplified in areas. Superlatives are used too often (e.g. interestingly, fascinatingly).
Answer: We tried to reduce the use of superlatives.
Reviewer 3 Report
In this manuscript, authors aim to investigate whether DLBCL cancer cells (cell line Riva) that have become resistant to venetoclax were more sensitive to BIRD-2. Therefore, they compared BIRD-2 sensitivity in an isogenic cell model with acquired venetoclax resistance. The results showed that Riva resistant cells, do not develop increased sensitivity towards BIRD-2 compared to their naïve counterparts. Consistently, not upregulation of the IP3R, was observed and authors suggested that Bcl-XL upregulation was, at least in part, responsible for the resistance to venetoclax.
The findings are interesting the manuscript is simple but l wrote directly and well presented.
Minor point
The authors concluded that induction of venetoclax resistance also results in resistance against BDA-366. Since, Bcl-2 protein levels are increased in Riva resistant cells, it is reasonable suggested that the induced cross-resistance towards BDA-366, observed in these cells; it was produced by the higher levels of Bcl-2 protein in the resistant cells. This point should be discussed in the manuscript.
Author Response
In this manuscript, authors aim to investigate whether DLBCL cancer cells (cell line Riva) that have become resistant to venetoclax were more sensitive to BIRD-2. Therefore, they compared BIRD-2 sensitivity in an isogenic cell model with acquired venetoclax resistance. The results showed that Riva resistant cells, do not develop increased sensitivity towards BIRD-2 compared to their naïve counterparts. Consistently, not upregulation of the IP3R, was observed and authors suggested that Bcl-XL upregulation was, at least in part, responsible for the resistance to venetoclax.
The findings are interesting the manuscript is simple but l wrote directly and well presented.
Answer: We thank the reviewer for his appreciation of the work.
Minor point
The authors concluded that induction of venetoclax resistance also results in resistance against BDA-366. Since, Bcl-2 protein levels are increased in Riva resistant cells, it is reasonable suggested that the induced cross-resistance towards BDA-366, observed in these cells; it was produced by the higher levels of Bcl-2 protein in the resistant cells. This point should be discussed in the manuscript.
Answer: We thank the reviewer for this suggestion. However, we did not find Bcl-2 upregulation. Furthermore, BDA-366 does not seem to be a bona fide inhibitor of Bcl-2, cfr. Villalobos-Ortiz M et al, Cell Death Differ, 2020. Finally, even if Bcl-2 would be upregulated, one would expect that cells become more sensitive to BDA-366 as it was proposed to convert Bcl-2 from an anti-apoptotic into a pro-apoptotic protein.
Reviewer 4 Report
Although BH3-mimetic drug venetoclax has been approved and is used for cancer treatment, cell tend to develop a resistance. In the manuscript one of the mechanism of resistance is described. Authors cultivated venetoclax sensitive cell line in the presence of drug to induce resistance and compared response of two lines (sensitive and resistant) to speceific inhibitors Bcl-2 family as well as several other cell line characteristics. They found the in case of the resistant line, the resistance was established by over expression of Bcl-XL (an antiapoptotic Bcl-2 family member, other than target of venetoclax Bcl-2).
I found these results relevant and significant. The design of the experiments is straightforward and experiments are well executed and described in the manuscript. Results and conclusions are appropriately discussed.
I therefore recommend to accept the manuscript for the publication in Biomolecules as it is.
Only one minor concern, in line 42 of the manuscript 'B-cell homology (BH) domains' should be corrected to 'Bcl-2 homology (BH) domains'.
Author Response
Although BH3-mimetic drug venetoclax has been approved and is used for cancer treatment, cell tend to develop a resistance. In the manuscript one of the mechanism of resistance is described. Authors cultivated venetoclax sensitive cell line in the presence of drug to induce resistance and compared response of two lines (sensitive and resistant) to speceific inhibitors Bcl-2 family as well as several other cell line characteristics. They found the in case of the resistant line, the resistance was established by over expression of Bcl-XL (an antiapoptotic Bcl-2 family member, other than target of venetoclax Bcl-2).
I found these results relevant and significant. The design of the experiments is straightforward and experiments are well executed and described in the manuscript. Results and conclusions are appropriately discussed.
I therefore recommend to accept the manuscript for the publication in Biomolecules as it is.
Only one minor concern, in line 42 of the manuscript 'B-cell homology (BH) domains' should be corrected to 'Bcl-2 homology (BH) domains'.
Answer: We thank the referee for the appreciation of our work. We corrected this statement.
Round 2
Reviewer 1 Report
I have no further comments to the current version of the manuscript
Author Response
We thank the reviewer for his/her feedback, which helped us to further improve our manuscript.
Reviewer 2 Report
Thank you for the extensive edits which provided a much stronger and easier to read manuscript.
Minor edits:
Figure 4 legend still reads "Figure 1"
Line 321, be careful using the word "proved" - yes the data support this hypothesis in one cell line model, perhaps different terminology would be better
Author Response
We thank the reviewer for his/her constructive comments, which helped us to improve the manuscript.
- We couldn't locate the writing of Figure 1 in the legend of Figure 4.
- We amended the statement.